# DEAD-Box Helicase 3 Modulates the Non-Coding RNA Pool in Ribonucleoprotein Condensates During Stress Granule Formation

**DOI:** 10.3390/ncrna11040059

**Published:** 2025-08-01

**Authors:** Elizaveta Korunova, B. Celia Cui, Hao Ji, Aliaksandra Sikirzhytskaya, Srestha Samaddar, Mengqian Chen, Vitali Sikirzhytski, Michael Shtutman

**Affiliations:** Department of Drug Discovery & Biomedical Sciences, College of Pharmacy, University of South Carolina Columbia, 715 Sumter St., Columbia, SC 29208, USA; celiacui@upenn.edu (B.C.C.); ji5@email.sc.edu (H.J.); sikirzha@mailbox.sc.edu (A.S.); srsamaddar17@gmail.com (S.S.); chen256@email.sc.edu (M.C.); sikirzhy@mailbox.sc.edu (V.S.)

**Keywords:** stress granules, P-bodies, DDX3, RK-33, non-coding RNA, ribonucleoprotein condensates

## Abstract

Stress granule formation is a type of liquid–liquid phase separation in the cytoplasm, leading to RNA–protein condensates that are associated with various cellular stress responses and implicated in numerous pathologies, including cancer, neurodegeneration, inflammation, and cellular senescence. One of the key components of mammalian stress granules is the DEAD-box RNA helicase DDX3, which unwinds RNA in an ATP-dependent manner. DDX3 is involved in multiple steps of RNA metabolism, facilitating gene transcription, splicing, and nuclear export and regulating cytoplasmic translation. In this study, we investigate the role of the RNA helicase DDX3’s enzymatic activity in shaping the RNA content of ribonucleoprotein (RNP) condensates formed during arsenite-induced stress by inhibiting DDX3 activity with RK-33, a small molecule previously shown to be effective in cancer clinical studies. Using the human osteosarcoma U2OS cell line, we purified the RNP granule fraction and performed RNA sequencing to assess changes in the RNA pool. Our results reveal that RK-33 treatment alters the composition of non-coding RNAs within the RNP granule fraction. We observed a DDX3-dependent increase in circular RNA (circRNA) content and alterations in the granule-associated intronic RNAs, suggesting a novel role for DDX3 in regulating the cytoplasmic redistribution of non-coding RNAs.

## 1. Introduction

Stress granules (SGs) are biomolecular condensates formed from mRNA and RNA-binding proteins (RBPs) in the cytoplasm through liquid–liquid phase separation—a reversible process that leads to the formation of liquid-like droplets in cytoplasm—in response to various cellular stresses, including UV irradiation, heat, osmotic stress, endoplasmic reticulum stress, oxidative stress, and viral infections [1]. While SGs have traditionally been associated with translational shut-off and mRNA storage during stress, emerging research has revealed a more complex functional landscape. Recent findings highlight several key aspects of SG activity: (a) a portion of mRNAs recruited to SGs displays translational activity, and translational repression has been shown as a process co-occurring with SG formation [2]; (b) SGs maintain cellular homeostasis by preventing cell death pathways [3,4,5] and delaying premature senescence [6]; and (c) SGs regulate cellular metabolism, including inhibiting fatty acid oxidation during starvation through direct interactions with mitochondria and lipid droplets [7]. SG dysregulation is implicated in various pathologies [8], including cancer, where increased SG formation and persistence contribute to resistance to cell death, and senescence, where defects in SG dynamics are observed [9]. In neurodegeneration, SGs may seed pathological protein aggregation, while impaired SG transport along axons disrupts mRNA trafficking to synapses [10]. SGs are also exploited by viruses, which manipulate SG-associated RBPs to enhance replication and alter stress responses [11].

The diverse functions and associated pathologies of SGs arise from their complex and heterogeneous organization. According to the conventional model, mature SGs consist of dense cores surrounded by less dense shells [12], both composed of mRNA and proteins, with dynamic exchange occurring between the core, shell, and surrounding cytoplasm [13]. Comparative studies reveal a conserved SG protein pool, including RBPs, prion-like proteins, translation factors, and ~50% uncharacterized proteins [12] probably diffused in SGs during their cytosol journey [14]. SG-associated mRNAs encode proteins for ATP binding, transcription, cell cycle regulation, and splicing [15]. Despite extensive data collection, significant gaps remain in understanding the biophysical mechanisms of SG assembly, organization, and dynamics [12,13]; transcriptome [15] and protein composition [12,14]; and interactions with other biomolecular condensates (e.g., P-bodies [14], Q-bodies [16]) and organelles [7], particularly across different cell types and stress conditions [17,18].

Using inhibitors that block glycolysis and oxidative phosphorylation in U2OS cells, it was shown that ATP is required for SG assembly, SG fusion, and the dynamic exchange of G3BP—a marker and one of the key drivers of SG formation—between SGs and the cytoplasm [12]. Later, it was reported that proteins from the RNA-dependent DEAD-box ATPase (DDX) family—which interact with RNA and contain low-complexity domains (LCDs, loosely structured protein regions that are essential for oligomerization and enable the formation of protein networks)—can modulate the assembly and disassembly of biomolecular condensates such as stress SGs and P-bodies [19,20]. The proposed mechanism suggests that the formation of complexes between DDX family proteins (such as yeast Dhh1 and Ded1), ATP, and RNA drives liquid–liquid phase separation, resulting in the formation of DDX–RNA condensates, as shown in vitro [20]. Moreover, the reverse process, driven by the activation of Ded1 ATPase activity through the C-terminal region of eIF4G, was shown to lead to the disassembly of these complexes.

Our previous work indicates that the inhibition of ATPase activity in human Dead-Box Helicase 3 (DDX3), the orthologue of yeast Dhh1 and Ded1 and a component specific to mammalian SGs, using RK-33 or the knockdown of DDX3 in U2OS cells, leads to a decrease in the rate of SG assembly but does not affect disassembly [21]. DDX3, an ATP-dependent helicase, localizes to both the nucleus and cytoplasm, with a higher nuclear concentration in healthy cells [22]. DDX3 has been shown to hydrolyze ATP to unwind RNA-RNA and DNA-RNA duplexes and to recruit RNA into SGs in mammalian cells through the formation of DDX3–RNA complexes and protein oligomerization [20,23]. However, DDX3 activity extends to nearly all steps of RNA metabolism, including nuclear mRNA transport, ribosomal and nuclear small RNA export, 5′ UTR translation activation, and RNA splicing, although its precise roles remain unclear [22]. Therefore, we hypothesize that alterations in DDX3 activity can influence not only the SG dynamics but also lead to changes in the RNA pool, both in RNP granules, including SG and associated condensates such as P-bodies, and the cytoplasm. Here, we report that the analysis of RNP granule content purified from U2OS cells treated with sodium arsenite—a known inducer of both SG and P-bodies [24]—and RK-33, a small molecule inhibitor of DDX3 ATPases activity shown to be effective in breast cancer bone metastasis patients [25], reveals alterations in the non-coding RNA pools of both RNP granules and the cytoplasm. Specifically, we observe an increase in intron RNA and circular RNA (circRNA) within RNP granules.

## 2. Results and Discussion

To investigate the impact of DDX3 activity on the RNA composition of RNP granules, we induced the formation of SGs and P-bodies in U2OS cells using sodium arsenite (NaAsO_2_), an inducer of oxidative stress [24]. The inhibition of DDX3 activity was achieved using the small molecule RK-33 before NaAsO_2_ treatment. RK-33 has been shown to specifically inhibit the ATPase and ATP-dependent RNA helicase activities of DDX3 by binding to its ATP-binding site [26,27], leading to decreased SG assembly as described in our previous work [21] (Figure 1A, Appendix A). Notably, a small subset of cells in the population exhibited stress granules even in the absence of induced stress. Following cell lysis, the RNP granule-enriched fraction was obtained through a series of centrifugation steps [28] (Figure 1B) from cells that were either untreated, treated with NaAsO_2_ or RK-33 alone, or co-treated with both compounds, as described in the Section 3. Western blot was used to validate the efficiency of the assay in isolating RNP-enriched stress granules, both under basal conditions and following NaAsO_2_ treatment. RNA was then purified from this fraction. The separation of G3BP1 between insoluble and soluble fractions corresponds to the behavior of G3BP1 as a canonical G3BP1 marker and repeats the DDX3 pattern in Western blot. Interestingly, it has been shown that G3BP1, while serving as a marker for stress granules, exhibits different behavior in its response to stress [29]. Recent publications have provided evidence that SGs are formed from pre-existing nanoscale seeds, which are present before the application of stress. The accumulation of both G3BP1 and DDX3 in SG-enriched fractions in both NaAsO_2_ naïve conditions pointed to the enrichment of cores [30,31,32]. The oxidative stress did not cause any further enrichment of G3BP1 and DDX3 in the SG-enriched fraction. This may suggest that most of these proteins are localized in nano-seeds, and SGs form by the redistribution of nano-seed proteins rather than the recruitment of soluble G3BP1 and DDX3. Additional extensive experiments are needed to confirm this. The RNA composition of the soluble and insoluble (SG- and P-body-enriched) fractions was determined after rRNA depletion using Illumina sequencing. The obtained reads were aligned to the genome, and the reads aligned to exons and introns were counted separately to determine processed mRNAs (or exonic) and intronic transcripts, which may represent intron-derived transcripts, lariat RNAs, or non-spliced mRNAs exported to the cytoplasm [33,34,35] (see the Section 3 for details). We observed an increase in intronic RNA levels between the soluble (~20%) and insoluble (~30%) cytoplasmic fractions across different conditions (Figure 1C). Moreover, we observed an increase in intronic RNA levels within the insoluble (RNP granule) fraction isolated from cells co-treated with RK-33 and NaAsO_2_, compared to the insoluble fraction from cells treated with NaAsO_2_ alone (Figure 1C, Appendix A). The differential expression analysis revealed a difference in RNA content between the soluble and insoluble fractions in NaAsO_2_-treated cells, as well as a significantly smaller difference between these fractions in cells co-treated with NaAsO_2_ and RK-33 (Figure 1D). The differential analysis of combined data from both the soluble and insoluble fractions revealed a decrease in cytoplasmic RNA content following co-treatment with NaAsO_2_ and RK-33. This observation is consistent with previous reports showing changes in the transcriptome landscape following RK-33 treatment [36,37] and, notably, shows further alterations between the RNP-enriched insoluble fraction and the soluble cytoplasmic fraction after RK-33 treatment. Further, co-treatment with RK-33 also resulted in the loss of the RNP-enriched fraction containing long non-coding RNAs (lncRNAs) such as NORAD—an lncRNA that regulates PUM protein activity and contributes to the formation of RNP condensates like stress granules [38]. Additional analysis revealed a higher intron-to-exon ratio in the insoluble RNP-enriched fraction compared to the soluble cytoplasmic fraction (Figure 1E, Appendix A).

Given the enrichment of non-coding intronic sequences, we analyzed the distribution of another class of ncRNAs, circRNAs. To determine circRNAs, we identified back-splice junctions in RNA sequencing data—a method that detects upstream exons appearing downstream, a characteristic feature of circRNAs (see the Section 3 for details). circRNA content was higher in the soluble fraction than in the insoluble fraction, associated with RNP granules, under all conditions (Figure 2A,B, Appendix A). As expected and consistent with previous reports, the overall expression of circRNAs remained relatively low [39]. Notably, NaAsO_2_ treatment led to an increase in circRNA content in the soluble fraction (Figure 2B), in line with the slow and inefficient biogenesis of circRNAs previously described [39]. Specifically, we observed an increase in circRNAs reported to be associated with proliferation or apoptosis signaling, linked to genes such as ARID1A [40,41], MPP6 [42], MAN1A2 [43,44], CREBBP [45,46], MGA [47], MAP3K4 [48], ZNF609 [49], and KANSL1 [50] (Figure 2A). Interestingly, the recently reported circMPP6 was found to interact with the RBP MEX3A [51], facilitating its association with P-body proteins, while ARID1A has been reported to interact with KHSRP [52,53], which can also be recruited into P-bodies. Both circMPP6 and circARID1A showed increased levels in the soluble fraction, accompanied by a decrease in the RNP granule fraction during NaAsO_2_ treatment. Subsequent treatment with RK-33 decreased circRNA content in the soluble fraction, and co-treatment with NaAsO_2_ and RK-33 prevented the stress-induced increase in circRNA content in the soluble fraction. Notably, in both cases, an increase in circRNA levels in the insoluble fraction was observed.

The molecular mechanism of DDX-dependent RNA redistribution is yet to be identified. A plausible explanation for the observed effects of RK-33, a DDX3 ATPase activity inhibitor, on the non-coding RNA pool is altered substrate preference and the RNA-binding efficiency of DDX3. To explore further, we analyze the distribution of DDX3 and the commonly used marker of SGs, G3BP1, in the NaAsO_2_-induced stress granules with expansion and super-resolution microscopy (Figure 2C,D and Figure 3). Strikingly, DDX3 forms distinct cores within SGs, which may or may not overlap with G3BP1 (Figure 2C,D). Moreover, STED super-resolution imaging suggested a reduction in stress granule size following co-treatment with RK-33 and NaAsO_2_, compared to NaAsO_2_ treatment alone (Figure 3). However, further quantitative analysis is required to confirm and quantify this observation. The results pointed out the possibility that DDX3 molecules form distinct domains in SGs, selectively enriched with specific RNAs. Recent reports have shown that DDX3 ATPase activity is required to enhance the dynamics of G3BP1-driven RNP granules, such as SGs, as well as to support ongoing translation in granules [54].

However, the activity of DDX3 is not restricted to RNP granules. The changes in intron and circRNA levels observed in our results (Figure 1 and Figure 2) may be linked to an alteration in DDX3’s role in splicing—a function that remains poorly characterized in the literature. Notably, the C-terminus of DDX3 contains a domain similar to those found in splicing factors [55], and DDX3 has been shown to interact with spliceosome and ribonucleoprotein complexes post-splicing [22]. Another potential explanation is that RK-33 alters DDX3’s role in nuclear export. DDX3 has been shown to interact with the TAP protein, which mediates the nuclear export of mRNA and may itself be transported via the TAP-dependent pathway [56]. Additionally, DDX3 has been shown to facilitate the export of intron-containing HIV-1 RNA through the CMR1-mediated nuclear export pathway [57]. Finally, DDX3 has also been reported to enhance transcription by interacting with transcription factors [58]. For example, circRNA *Circ-CTNNB1* binds to DDX3, facilitating its interaction with the transcription factor Yin Yang 1, which leads to β-catenin activation [59]. Given the numerous activities of DDX3 in RNA processing and localization, further analysis is needed to dissect the effects.

Overall, our results are the first to determine the function of DDX3 activity in regulating the subcellular distribution of circRNAs and intron-associated RNAs in SG- and P-body-enriched fractions. Importantly, the SGs’ core structure persists in both naïve and stressed conditions; therefore, the results demonstrate the DDX3 dependence of the RNA compositions of seed nanocores and stress-induced stress granules. Interestingly, co-treatment with RK-33 and NaAsO_2_ led to the depletion of non-coding RNAs—such as lncRNA NORAD [38], circMPP6 [42], and ARID1A [40]—that promote RNP granule formation or interact with associated RBPs (Figure 1D and Figure 2A). This highlights DDX3 activity as a potential target for investigating the role of non-coding RNAs—especially circRNAs, whose involvement in regulating RNP granule formation was recently suggested [60]. Notably, RK-33 co-treatment with NaAsO_2_ decreased stress response RNAs—such as ATF3 [61], DNAJB1 [62], HSPA6 [63], and HSPA1B [64]—compared to arsenite alone, with reductions in ATF3, DNAJB1, and certain circRNAs associated with cancer corresponding to RK-33’s established anti-tumor effects [25]. However, our preliminary results do not provide a definitive answer regarding the specific mechanism by which DDX3 activity influences non-coding RNA content in RNP granules and their formation. This opens several avenues for future investigation, including the identification of RNAs that directly interact with DDX3—potentially using RNA editing enzyme fusions—as well as the application of FISH techniques to validate the localization [15,65] depletion of lncRNAs and circRNAs within RNP granules. In addition, testing different RK-33 pre-treatment time points prior to stress induction may help elucidate the temporal dynamics of its effects on the non-coding RNA pool, particularly cytoplasmic circRNAs, whose biogenesis has been reported to be inefficient [39].

## 3. Methods

### 3.1. Cell Culture and Stress Granule Induction

U2OS cells were maintained in high-glucose Dulbecco’s Modified Eagle Medium (ATCC, Manassas, VA, USA) supplemented with 10% fetal bovine serum (Mediatech Inc., Manassas, VA, USA) and 1% penicillin/streptomycin (HyClone, GE Healthcare Life Sciences, Pittsburgh, PA, USA) at 37 °C in a humidified atmosphere with 5% CO_2_, as previously described [21]. Cells were pre-treated with 6 μM RK-33 (MedChemExpress, Monmouth Junction, NJ, USA) (, stock solution 5mM DMSO (MilliporeSigma Burlington, MA, USA D8418) for 1 h. Sodium arsenite stress was induced by treating cells with sodium arsenite, 500 µM NaAsO_2_ (MilliporeSigma, Burlington, MA, USA S7400), for 60 min. Basal conditions were defined as treatment with 0.12% DMSO for 2 h, corresponding to the duration and timing of RK-33 addition in treated samples.

### 3.2. Expansion Microscopy

The expansion microscopy procedure was performed based on previously published work by Boyden’s laboratory, with some modifications [66]. Briefly, U2OS cells treated with NaAsO_2_ were fixed for 5 min at room temperature using a fixative solution containing PFA and Triton X-100, followed by an additional permeabilization step with 0.5% Triton X-100. Coverslips with the cells were carefully washed for 10 min in a sodium borohydride solution to reduce autofluorescence and potentially restore antigenicity after fixation. Cells were then blocked with 5% BSA at room temperature for one hour.

The primary antibody mixture—mouse anti-DDX3 (sc-365768, Lot# D0819, 200 µg/mL, Santa Cruz Biothechnology, Dallas, TX, USA) and rabbit anti-G3BP1 (A302-033A-T, Lot# 2, 200 µg/mL, Fortis Life Sciences, Boston, MA, USA)—was applied overnight at 4 °C. Secondary antibodies used for visualization were Abberior StarRed 639–conjugated anti-mouse for DDX3 and donkey anti-rabbit Alexa Fluor Plus 555 for G3BP1.

For anchoring, coverslips were pre-incubated with 100 mM sodium bicarbonate twice for 15 min each, followed by incubation in 0.1% (*w*/*v*) GMA in 100 mM sodium bicarbonate for 3 h at room temperature, and then transferred for overnight incubation at 37 °C.

The gelation procedure was performed in a cold room (4 °C) to prevent premature gelation. The gelling chamber was assembled according to the referenced protocol [66]. Samples were carefully submerged in a mixture composed of StockX (8.6% (*w*/*v*) sodium acrylate (SA), 2.5% (*w*/*v*) acrylamide (AA), 0.15% (*w*/*v*) N,N′-methylenebisacrylamide (Bis), 2 M sodium chloride (NaCl), and 1× PBS, with 10% (*w*/*v*) N,N,N′,N′-Tetramethylethylenediamine (TEMED) and 10% (*w*/*v*)) ammonium persulfate (APS) in a 47:1:1:1 ratio. The samples were left at 4 °C for 30 min.

Gelled samples were transferred into a humidified tip box and incubated at 37 °C for 2 h. After gelation, gels were trimmed and incubated overnight in digestion buffer consisting of 8 U/mL Proteinase K (ThermoFisher Scientific, Waltham, MA, USA), Catalog No. BP1700-100), 0.5% (*w*/*v*) Triton X-100, 1 mM EDTA, 2 M NaCl, and ddH_2_O.

The following day, coverslips were placed in 10 mL of ddH_2_O in Petri dishes and shaken at 60 rpm on an orbital shaker for 10 min at room temperature, in the dark (protected with aluminum foil). This washing step was repeated twice. Finally, the samples were trimmed to fit into glass-bottom 35 mm Petri dishes for image acquisition.

Microscopic data were acquired using a Carl Zeiss LSM 700 laser scanning confocal microscope (Carl Zeiss Meditec, Dublin, CA, USA) equipped with a 20× Plan APO 0.8 air objective. Imaging was performed in Z-stack mode with a total of 17 optical sections. Images were captured at 1× scanning zoom with a resolution of 20.15 pixels per micron in 16-bit depth. Excitation was performed at 555 nm and 639 nm wavelengths. Subsequent image processing and analysis were conducted using ImageJ 1.54p software (National Institutes of Health, USA).

### 3.3. STED and Confocal Microscopy

Stimulated Emission Depletion (STED) microscopy was performed using a Stedycon STED microscope on fixed U2OS cells treated with 500 μM NaAsO2 for 60 min and labeled for stress granule markers. G3BP1 was detected with Abberior STAR RED and DDX3 with Abberior STAR ORANGE secondary antibodies. Images were acquired at a pixel size of 30 nm, with line accumulations of 10 for STED and 5 for confocal channels. Pixel dwell times were 10 µs (STED) and 5 µs (confocal). Excitation lasers were 640 nm (STAR RED) and 561 nm (STAR ORANGE), with STED at 775 nm (28.1% for STAR RED, 50% for STAR ORANGE). Gated detection was used for STED (1–7 ns), while confocal channels had no gating. Images were acquired with a 100× objective, using a 32 µm pinhole, over a field of view of 34 × 63 µm (1123 × 2087 pixels).

Confocal microscopy (40×) and STED super-resolution microscopy (100×) were also performed using the STELLARIS 8 STED system on samples treated with NaAsO_2_, RK-33, or a combination of both, with concentrations as described above and 1% DMSO as the basal condition. G3BP1 was detected with Abberior STAR RED, DDX3—with Aberrior STAR ORANGE. Excitation lasers were 642 nm (STAR RED) and 589 nm (STAR ORANGE), with STED at 775 nm (80% for STAR RED, 90% for STAR ORANGE). Gated detection was used for STED STAR RED (1–7 ns), STAR ORANGE in Intensity mode, while confocal channels had no gating. Confocal microscopy (20x) of these samples was performed on a Zeiss LSM 700 microscope. Images were acquired at 2048 × 2048 resolution with a pixel size of 0.3126 μm. Detection was performed with PMT detectors, using 555 nm and 639 nm excitation lasers.

### 3.4. Stress Granule (SG) Isolation

Stress granule isolation was adapted from previous protocols [28,67]. Briefly, cells were grown in 20 mL culture dishes, washed once with ice-cold culture medium and once with ice-cold PBS, then scraped and collected by centrifugation at 200× *g* for 5 min at 4 °C. Pellets were washed with 500 µL ice-cold PBS, centrifuged at 300× *g*, and lysed in buffer [50 mM Tris (pH 7.6), 50 mM NaCl, 5 mM MgCl_2_, 0.5% NP-40 (MilliporeSigma Burlington, MA, USA 492016, Lot B21198), 1 mM β-mercaptoethanol (Bio-Rad, Hercules, CA, USA #1610710), 1× EDTA-free protease inhibitor cocktail (Roche, Indianapolis, IN, USA, 05 892 791 001), and 0.4 U/µL RNase inhibitor (New England Biolabs, Ipswich, MA, USA M0314L)]. Lysates were homogenized with 30 strokes using a Dounce homogenizer and centrifuged at 300× *g* for 3 min at 4 °C to remove nuclei. Then the supernatant was centrifugated at 10,000× *g* for 10 min at 4 °C. The soluble and insoluble (RNP granule) cytoplasmic fractions were separated, and protein and RNA concentrations were measured using a NanoDrop spectrophotometer.

### 3.5. Western Blot

After protein concentrations were measured using a NanoDrop spectrophotometer, as described in the Section 3.4, equal amounts of protein (approximately 80 μg from both soluble and insoluble fractions) were loaded onto SurePAGE 4–12% Bis-Tris gradient gels. Following electrophoretic separation, proteins were transferred onto PVDF membranes (Cat# 88518, Thermo Fisher Scientific, Waltham, MA, USA). The membranes were blocked in 1× TBS-T containing 5% non-fat dry milk and incubated overnight at 4 °C with primary antibodies. The primary antibodies applied included anti-G3BP1 (1:1000, A302-033A, Thermo Fisher Scientific), anti-DDX3 (1:3000, sc-365768, Santa Cruz Biotechnology, Dallas, TX, USA), and anti-GAPDH (1:5000, #2118, Cell Signaling Technology). HRP-conjugated secondary antibodies, anti-mouse (1:8000, #7076S, Cell Signaling Technology, Danvers, MA, USA), and anti-rabbit (1:8000, #7074S, Cell Signaling Technology, Danvers, MA) were used for detection. Protein bands were visualized using the SuperSignal West Femto Maximum Sensitivity Substrate (ThermoFisher Scientific).

### 3.6. Intron and Circular RNA Analysis

RNA and library preparation, the post-processing of the raw data, and data analysis were performed by the USC CTT COBRE Functional Genomics Core. RNAs were extracted with Zymo Direct-zol RNA Microprep Kits as per manufacturer recommendations (Zymo, Tustin, CA, USA), and RNA quality was evaluated on RNA-1000 chip using Bioanalyzer (Agilent, Santa Clara, CA, USA). RNA libraries were prepared using an established protocol with NEBNext Ultra II Directional Library Prep Kit (NEB, Lynn, MA, USA). Each library was made with one of the TruSeq barcode index sequences and had a 1:200 dilution of Spike-in SIRV_Set3 (Lexogen, Greenland, NH, USA) added. Illumina sequencing was performed by Medgenome (Foster City, CA, USA) with Illumina HiSeqX (150 bp, pair-ended).

Human genome GRCh38.p13 (GRCh38_r98.all, GENCODE release 40) and comprehensive gene annotation GTF (Regions CHR, GENCODE release 40) were downloaded from GENCODE.

Sequences were aligned to the human genome GRCh38.p13 (GRCh38_r98.all, GENCODE release 40) using STAR 2.7.2b [68]. SAMtools (v1.2) was used to convert aligned SAM files to BAM files, and reads were counted using the feature-Counts function of the Subreads package [69] using a homemade gencode_v40annotation_intron.gtf annotation file. CircRNA was detected from chimeric reads by DCC 0.5.0 [70], the tool identifying circRNA via back-splice junctions.

A comprehensive gene annotation GTF was downloaded (Regions CHR, GENCODE release 40) from GENCODE. Using bedtools 2.27.1 for a homemade annotation file, first use mergeBed to merge all exons. Then, subtractBed was used to subtract the exonic region from the genic region to define intronic regions.

Only reads that were mapped uniquely to the genome were used for gene expression analysis. Differential expression analysis was performed in R using the edgeR package [71]. The average read depth for the samples was 18 million reads, and only genes with at least one count per million average depth were considered for differential expression analysis. Raw counts were normalized using the Trimmed Mean of M-values (TMM) method, and the normalized read counts were then fitted to a generalized linear model using the function glmFit [72]. Genewise tests for significant differential expression were performed using the function glmLRT. The *p*-value was then corrected for testing using Benjamini–Hochberg’s FDR [73].

To compare expression levels between soluble and insoluble fractions within the same treatment condition, a paired *t*-test was used [74]. For comparisons between treatment conditions, an unpaired *t*-test was applied. The experiment with RNA sequencing of soluble and insoluble fractions was repeated three times for all experiment conditions.

## 4. Conclusions

Here, we have demonstrated that the treatment of U2OS cells with RK-33, an inhibitor of DDX3 ATPase activity, leads to a redistribution of non-coding RNAs within RNP granules and the cytoplasm. Specifically, we observed a depletion of RNA species in SGs, including the lncRNA NORAD, which is known to support RNP condensate formation. We also showed that RK-33 treatment prior to SG induction with NaAsO_2_ prevented the stress-induced accumulation of circRNAs associated with proliferation and apoptosis signaling in the soluble fraction. Notably, RK-33 treatment prior to SG induction led to the depletion of circRNAs such as circMPP6 and ARID1A in the soluble cytoplasmic fraction, both of which have been reported to interact with RBPs associated with P-bodies and SGs. These findings are consistent with the reduced SG assembly we previously reported and are further supported by representative confocal images showing reduced SG formation.

These results highlight the potential role of DDX3 in modulating the non-coding RNA landscape, particularly regulatory RNAs involved in RNP granule formation and signaling-related circRNAs. Interestingly, we also noted a tendency toward increased levels of circRNA and intronic RNA within RNP granules post-treatment. While preliminary, these trends warrant further investigation. Future studies could explore longer RK-33 incubation times to assess RNA dynamics more fully, compare responses in non-cancerous cell lines to distinguish cancer-specific effects, and incorporate FISH-based validation of affected RNA species in granules.

## Figures and Tables

**Figure 1 ncrna-11-00059-f001:**
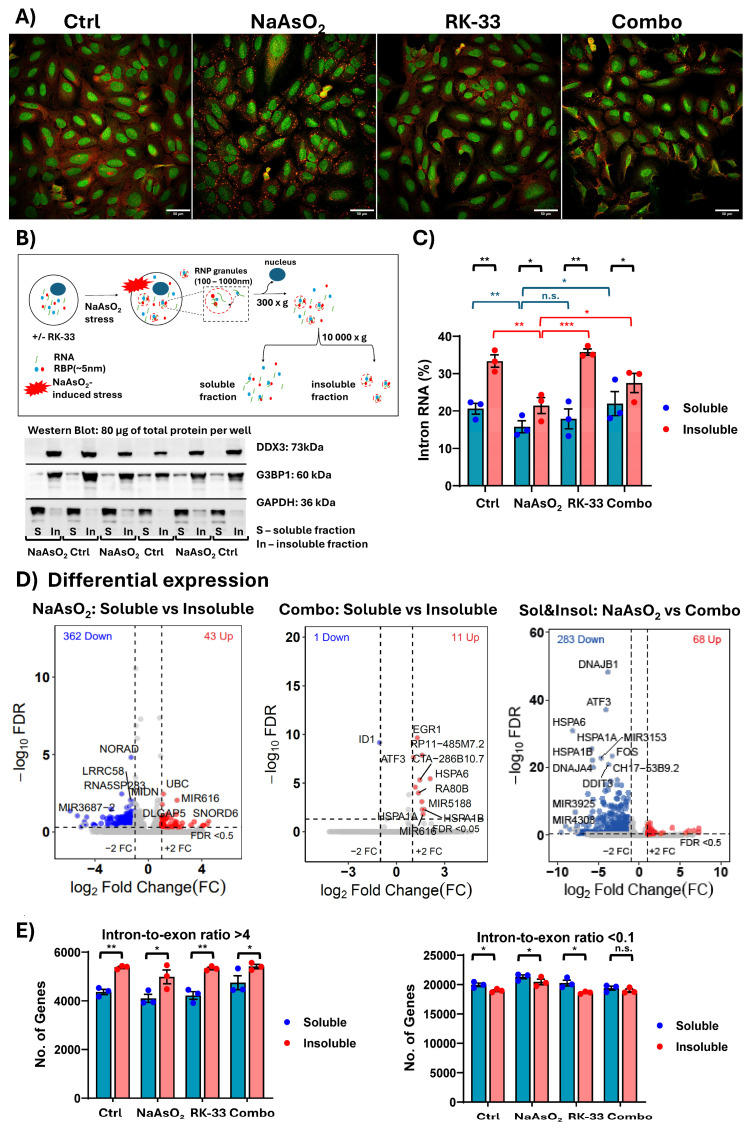
(**A**) Confocal images (40×) of U2OS cells immunostained for G3BP1 (STAR RED, red) and DDX3 (STAR ORANGE, green). Yellow regions indicate areas of colocalization between the two proteins. The picture demonstrates the formation of SG under four different conditions. (**B**) Schematic of RNA granule isolation: Cells were lysed, and the nuclear fraction was removed by centrifugation at 300× *g*. The cytoplasmic fraction was then separated into soluble and insoluble fractions by centrifugation at 10,000× *g*. Insoluble fractions were enriched for RNP granules such as SGs and P-bodies, based on the previous literature. Western blotting was performed in three replicas to verify the enrichment of SG markers in the corresponding fractions prior to sequencing analyses. (**C**) Level of intron RNA in soluble and insoluble fractions was estimated as the percentage of intron reads relative to total reads. (**D**) Differential analysis of the whole RNA content normalized to spike-in control between soluble and insoluble fractions: (**left**) cells treated with NaAsO_2_; (**middle**) cells co-treated with RK-33 and NaAsO_2_; (**right**) differential analysis of the whole RNA normalized to spike-in control dataset combined from soluble and insoluble fractions between cells treated with NaAsO_2_ and co-treated with RK-33 and NaAsO_2_. Reads-log_10_(FDR) < 0.5 and log_2_ (FC) > 2 were estimated as significant. FDR—false discovery rate, FC—fold change. (**E**) For each gene, the intron-to-exon ratio was estimated in both soluble and insoluble fractions, focusing on genes with ratios greater than 4 (**left**) and less than 0.1 (**right**). (**C**,**E**) Statistical comparisons between soluble and insoluble fractions within the same condition were performed using a paired *t*-test. Unpaired *t*-tests were used to compare soluble (blue) or insoluble (red) fractions between the NaAsO_2_ condition and other conditions across different experiments. *p*-value < 0.005 (***), *p*-value < 0.05 (**), *p*-value < 0.5 (*), *p*-value > 0.5 (n.s.) (**A**,**C**,**E**) Conditions: Ctrl–U2OS cell line without treatment; NaAsO_2_–U2OS cell line treated with sodium arsenite to cause stress granule formation; RK-33–U2OS cell line treated with RK-33, inhibitor of DDX3; Combo–U2OS cell line treated with RK-33 and sodium arsenite.

**Figure 2 ncrna-11-00059-f002:**
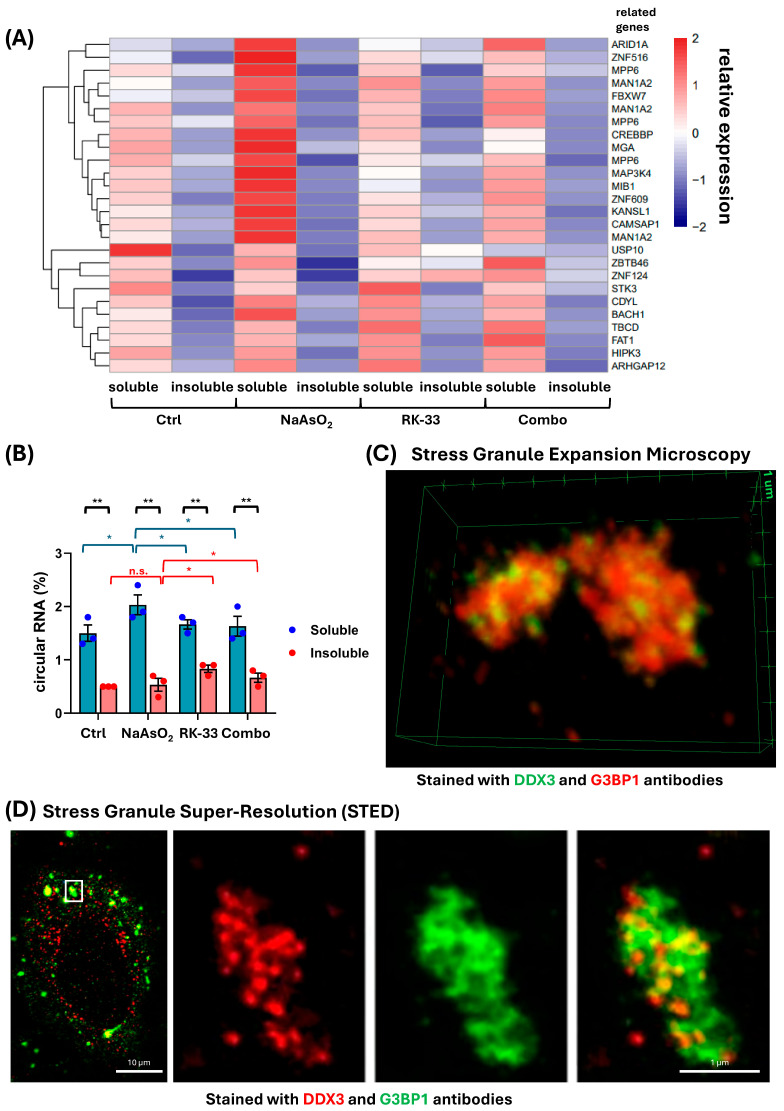
(**A**) Heatmap of circRNA levels in soluble and insoluble fractions under different treatment conditions; (**B**) level of circRNA in soluble and insoluble fractions was estimated as the percentage of circRNA reads relative to total reads. Statistical comparisons between soluble and insoluble fractions within the same condition were performed using a paired *t*-test. Unpaired *t*-tests were used to compare soluble (blue) or insoluble (red) fractions between the NaAsO_2_ condition and other conditions across different experiments. *p*-value < 0.05 (**), *p*-value < 0.5 (*), *p*-value > 0.5 (n.s.) Conditions: Ctrl–U2OS cell line without treatment; NaAsO_2_–U2OS cell line treated with sodium arsenite to cause stress granule formation; RK-33–U2OS cell line treated with RK-33, inhibitor of DDX3; Combo–U2OS cell line treated with RK-33 and sodium arsenite. (**C**) Three-dimensional image from expansion microscopy showing SGs in U2OS stained with DDX3 (STAR REd, green) and G3BP1 (Alexa Fluor Plus 555, red) antibodies. (**D**) STED super-resolution images of a U2OS cell stained with antibodies against DDX3 (STAR RED, red) and G3BP1 (STAR ORANGE, green). Insets show magnified views of stress granules (SGs) with individual channels for DDX3 and G3BP1, as well as the merged image. (**C**,**D**) SG formation was caused by treatment with NaAsO_2_. Yellow corresponds to the colocalization of DDX3 and G3BP1.

**Figure 3 ncrna-11-00059-f003:**
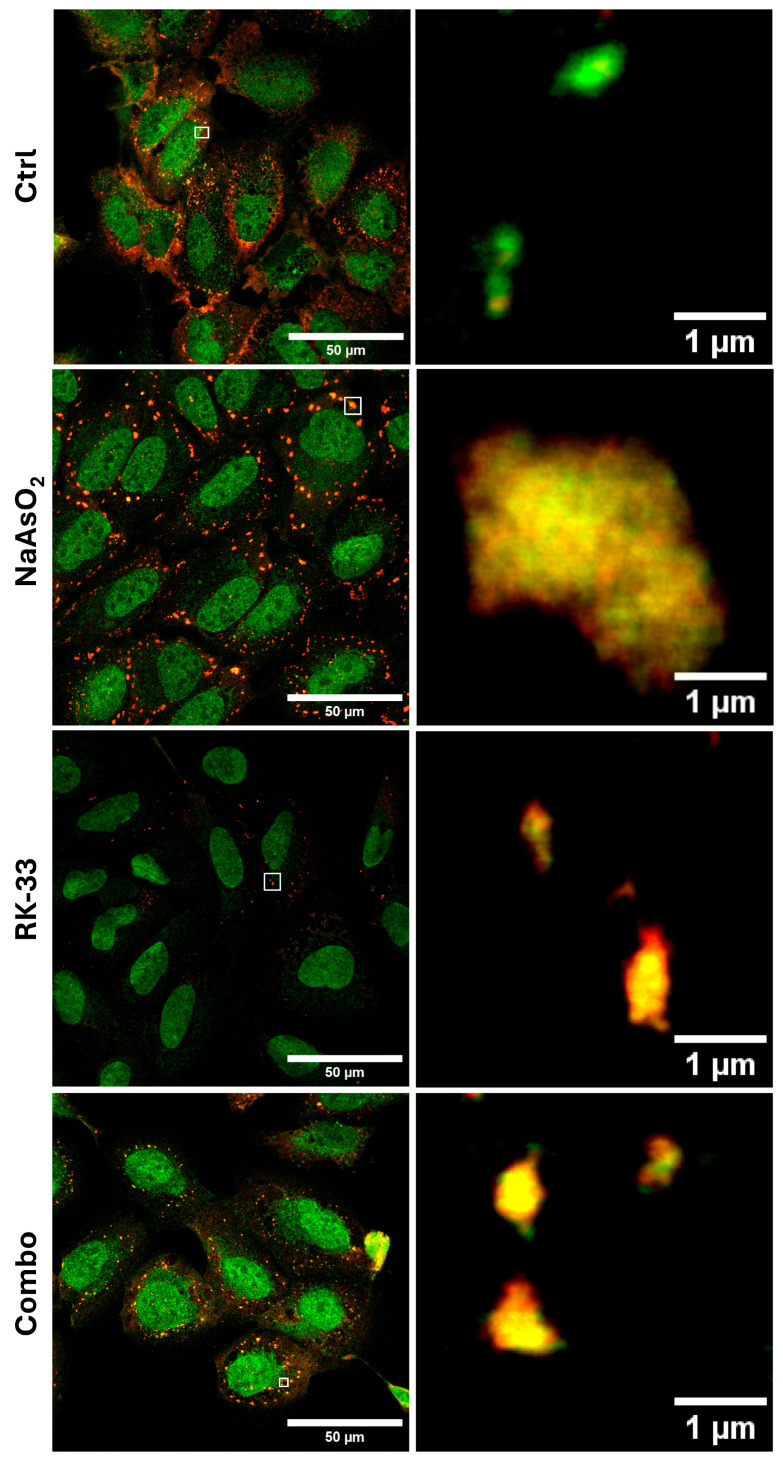
STED images (100×) of U2OS cells exhibiting SG formation, immunostained for G3BP1 (STAR RED, red channel) and DDX3 (STAR ORANGE, green channel). Yellow regions indicate areas of colocalization between the two proteins. White square is associated with the corresponding SG shown on the right. Conditions: Ctrl–U2OS cell line without treatment; NaAsO2–U2OS cell line treated with sodium arsenite to cause stress granule formation; RK-33–U2OS cell line treated with RK-33, inhibitor of DDX3; Combo–U2OS cell line treated with RK-33 and sodium arsenite.

## Data Availability

RNA sequencing data were deposited in the NCBI. GEO Accession number is GSE303136.

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
