# Peer review of "DEAD-Box Helicase 3 Modulates the Non-Coding RNA Pool in Ribonucleoprotein Condensates During Stress Granule Formation"

_ncrna, 2025, doi:10.3390/ncrna11040059_

Round 1

Reviewer 1 Report

Comments and Suggestions for Authors

The authors presented a short paper on the involvement of the DDX3 helicase in the distribution of mRNA exons and introns, and also circRNA, between the soluble fraction of the cell extract and the insoluble fraction in stress granules. In addition, the authors examined changes in the distribution between the fractions in the presence of the inhibitor of this helicase PK-33. The article has a very good and detailed introduction to the essence of the issue. The methods are described in great detail, which makes a pleasant impression. The results are well illustrated and discussed in detail. Statistical processing of the results looks reliable. The authors focus on the features of the distribution of statistics relative to the data obtained. The conclusions are formulated somewhat concisely, but correspond to the results obtained. No significant shortcomings were found in the article. The only recommendation is to more clearly separate the conclusion from the main material.

The main question of the study was formulated rather ambiguously. It can be best conveyed by a phrase from the abstract: «we investigate the role of the RNA helicase DDX3's enzymatic activity in shaping the RNA content of ribonucleoprotein (RNP) condensates». I cannot say that the obtained results provide an answer on this question. In fact, the authors attempted to investigate the effect of DDX3 activity on RNA composition of SG and the difference in RNA localization in the soluble and insoluble (SG) fractions after arsenate induced stress. 

It can be assumed that the approach used was quite original for studying the distribution of mRNA introns and circular RNA between the soluble and insoluble fractions depending on the activity of DDX3 and the effect of the enzyme inhibitor on this distribution. I cannot say whether this approach solves a specific gap in this area of research. How important the result is for solving a particular problem? Probably yes.

The results presented in Figure 2B showed weak differences between soluble and insoluble fractions in all treatments. The same effect can be found for the circRNA distribution (Figure 3B). It would be good if the authors provided more explanation for this small difference. Was it expected or not?
It will be interesting if the authors say something about difference in RNA content between the soluble and insoluble fractions in NaAsO2-treated cells (Fig. 1C). Is there any connection between stress and the nature of the mRNA that is expressed at that moment?
Figure 2B is rather useless to readers due to the scale of the image.

The main problem with the paper is that there is no structured conclusion about the work. It is unclear what has been done? There is no answer to the questions posed. What is the modulation of DDX3 for non-coding RNA if the difference between the soluble and insoluble fraction depending on the treatment seems small?

Reviewer 2 Report

Comments and Suggestions for Authors

The topic of the manuscript is interesting and essentially follows a recent article by the corresponding author published in Biochem Pharmacol, 371 2020. 182: p. 114280.

Some hypotheses are unique and the authors discuss them very reasonably. The relationship between DDX3 and ncRNA in SG under stress conditions has only been studied in the last few years. The authors' results are therefore very topical.

I have no serious comments on the present work. My comments are more of a technical concern and are related to the presentation of the data. The authors in the paper use a combination of software analyses that are difficult to verify by reading the manuscript and therefore these results could be described in more detail.

Comments

P3 L109            What was the concentration of NaAsO₂?

P3 L124            Citation to the refence protocol is missing.

P3 L142            Instead of Diode555 and Diode639 I recommend to use another

                        formulation; to write directly the wavelengths

Many typos: e.g. L176, 183, 211, 220, 231 etc. - 1:200dilute, featureCounts, cleft[34, 35].....

I would also mark the place of adding RK-33 and improve the resolution on the scheme Fig. 1a.

Image 2b is not representative enough, I would definitely not use direct software output, but process the data and especially improve the resolution again, it is not basically legible.

Conclusion

The manuscript needs a few edits to make it suitable for publication.

Reviewer 3 Report

Comments and Suggestions for Authors

Korunova et al hand in their work entitled "DEAD-box helicase 3 modulates the non-coding RNA pool in ribonucleoprotein condensates during stress granule formation". They describe an approach to investigate changes in the RNA content of stress granules upon inhibition of DEAD-box helicase 3/DDX3. By using a small molecule to inhibit DDX3 activity and inducing stress granule formation by sodium arsenite treatment, the authors aim to analyse the changes in content of SG by RNA sequencing. The results show an increase of circular RNA content and intron-associated RNAs in stress granules upon DDX3 inhibition.

The work is certainly novel and technically sound. I therefore see that the paper can be published in non-coding RNA. I suggest to take care of several of issues prior to publication.

Major issues:

The reader is left behind with the observation that circular RNAs enrich in SG when DDX3 is inhibited. Fair enough as a key observation, but what do we learn from this result? In other words: what is the significance of the finding? Do we know of special stress situations in which this may be relevant? Some ideas should be discussed.

Figure 1 jumps right into the comprehensive RNA analysis without giving insights into the characteristics of SG and SG markers under die conditions used: Immunofluorescence images of the cells after treatments (NaAs only, RK-33 only, combo) would help to make the point and address if/how SG form at all after treatment with RK-33. A Western Blot is required showing protein levels of purifications upon each treatment to confirm the enrichment of SG proteins (e.g. G3BP/DDX3) to accompany the RNAs.

Several issues also come with Figure 3: Can any of the identified RNAs be directly visualized within SG, i.e. by costaining with DDX3, (e.g.using FISH) to confirm the shortlist/heatmap ? Is there a significant increase of circular RNA content in the insoluble fraction already with RK-33 only? What is the explanation for this increase, as there should be no SG after this treatment only? Why not comparing  expanded SG with and without RK-33?

Minor issues:

  • Fig. 1: How are soluble/insoluble fractions further processed? (Supernatant vs. pellet?)
  • Fig. 1: and following: Statistical comparisons would be better visible above each other
  • Abbreviation for sodium arsenite is not consistent throughout the paper (NaAsO2 line 255 /NaAs line 249)
  • Abbreviation for control is not consistent throughout the paper (cntrl line 248/ ctrl in figures)
  • Figure 3 C) and D) description of stainings in images are unclear/missing.

Round 2

Reviewer 3 Report

Comments and Suggestions for Authors

Korunova et al hand in a quickly revised version of their manuscript "DEAD-box helicase 3 modulates the non-coding RNA pool in ribonucleoprotein condensates during stress granule formation". I do appreciate the efforts of the authors to better explain the outcome of their results (reformulated discussion). The set up to start with is easier to recapitulate with the IF images shown in Fig. 1A. However, I still do not find the information given for the fractions generated by sedimentation very insightful. A Western Blot is inserted that shows a non-consistent pattern of G3BP1 as a marker for SG formation. Apart from the fact that the blot misses essential labelling of the conditions analyzed, there is no correlation between SG formation and G3BP1 sedimentation. Citing Namkoong et al. doesn’t help as, there, SG sediments are very different between control and non-stressed cells. This remains an essential point to be clarified. Sedimentation under the different conditions and the correlated analysis of RNAs in the sediments is the basis for the biological relevance of the observations provided here. This issue needs to be solved. How can the pattern of G3BP1 be explained? Is G3BP1 just a bad marker for what is done here, or is the blot shown here just a bad example?

Round 3

Reviewer 3 Report

Comments and Suggestions for Authors

To my opinion, the revised manuscript of Korunova et al. now conveys the message well and explains procedure and biological conclusions to make the story complete.